# Myco–Phycobiont Interactions within the “*Ramalina farinacea* Group”: A Geographical Survey over Europe and Macaronesia

**DOI:** 10.3390/jof10030206

**Published:** 2024-03-08

**Authors:** Patricia Moya, Salvador Chiva, Tamara Pazos, Eva Barreno, Pedro Carrasco, Lucia Muggia, Isaac Garrido-Benavent

**Affiliations:** 1Instituto Cavanilles de Biodiversidad y Biología Evolutiva (ICBiBE)—Departament de Botànica, Universitat de València, C/Dr. Moliner, 50, E-46100 Burjassot, València, Spain; salvador.chiva@uv.es (S.C.); tamara.pazos@uv.es (T.P.); eva.barreno@uv.es (E.B.); 2Department of Life Sciences, University of Trieste, Via L. Giorgieri 10, 34127 Trieste, Italy; lucia_muggia@hotmail.com; 3Instituto de Biotecnología y Biomedicina (BIOTECMED), Universitat de València, E-46100 Burjassot, Spain; pedro.carrasco@uv.es; 4Departament de Botànica i Geologia, Universitat de València, C/Dr. Moliner, 50, E-46100 Burjassot, València, Spain; isaac.garrido@uv.es

**Keywords:** *Ascomycota*, climatic niche, haplotype, lichenized fungi, Macaronesia, microalgae, *Trebouxia jamesii*, *Trebouxia lynnae*

## Abstract

*Ramalina farinacea* is a widely distributed epiphytic lichen from the Macaronesian archipelagos to Mediterranean and Boreal Europe. Previous studies have indicated a specific association between *R. farinacea* and *Trebouxia* microalgae species. Here, we examined the symbiotic interactions in this lichen and its closest allies (the so-called “*R. farinacea* group”) across ten biogeographic subregions, spanning diverse macroclimates, analyzing the climatic niche of the primary phycobionts, and discussing the specificity of these associations across the studied area. The most common phycobionts in the “*R. farinacea* group” were *T. jamesii* and *T. lynnae*, which showed a preference for continentality and insularity, respectively. The Canarian endemic *R. alisiosae* associated exclusively with *T. lynnae*, while the other *Ramalina* mycobionts interacted with both microalgae. The two phycobionts exhibited extensive niche overlap in an area encompassing Mediterranean, temperate Europe, and Macaronesian localities. However, *T. jamesii* occurred in more diverse climate types, whereas *T. lynnae* preferred warmer and more humid climates, often close to the sea, which could be related to its tolerance to salinity. With the geographical perspective gained in this study, it was possible to show how the association with different phycobionts may shape the ecological adaptation of lichen symbioses.

## 1. Introduction

The lichen-forming fungus *Ramalina farinacea* (L.) Ach. is an iconic member of the family *Ramalinaceae* (*Lecanorales*, *Ascomycota*) [1] that develops pendant, whitish–greenish fruticose thalli composed of slender thallus ramifications (i.e., laciniae), which usually display well-developed asexual reproduction structures, i.e., soralia. These structures consist of soredia, minute clusters of fungal hyphae enwrapping microalgal cells, dispersed together by wind or animals [2,3]. Sexual reproduction involving meiotic spores produced in fungal apothecia is, however, exceptionally rare in this lichen. *Ramalina farinacea* is widely distributed across North America, Macaronesia, and Europe, from the climatically mild Mediterranean Basin to the cooler Boreal region [4], showing remarkable ecological versatility as it thrives in diverse habitats from tree trunks and branches in shaded deciduous forests to isolated trees exposed to sun and wind, as well as hedgerows, scrublands, rocks, and walls [5]. Although this lichen is relatively easily identifiable in the field, its populations also show variability in morphology and chemical composition, producing various secondary metabolites [6]. That variability was the basis for the description of the related species *R. subfarinacea* (Nyl. ex Cromb.) Nyl. and *R. alisiosae* by Pérez-Vargas and Pérez-Ortega [7,8]. In a recent paper, Moya et al. [4] showed that all these species together with an undescribed *Ramalina* sp. are indeed phylogenetically closely related, as they likely evolved recently and therefore still represent a species complex, the so-called “*Ramalina farinacea* group”. Indeed, phylogenetic analyses using a pair of genetic markers, including the fungal barcode nuclear ribosomal internal transcribed spacer (nrITS), suggested incomplete lineage sorting and conflicting phylogenetic signal [4].

The work by Moya et al. [4] also studied the phylogeography of the fungal partners (mycobionts) in the *Ramalina farinacea* group. In particular, they evaluated the hypothesis proposed by Krog and Østhagen [9], posing an origin for *R. farinacea* in the Macaronesian–Mediterranean region and its gradual geographic expansion to most temperate and Boreal regions of the Northern Hemisphere, with the Canary Islands as its probable southernmost limit in the Atlantic region. Aptroot and Schumm [10] have extended this boundary to include Cape Verde, another Macaronesian archipelago south of the Canary Islands. Preliminary phylogenetic studies by Del Campo et al. [11] and Molins et al. [12] had identified several clades in the mycobiont phylogeny, some with specimens restricted to the Iberian Peninsula or the Canary Islands, whereas at least one clade was found to be shared among these two regions and California. Finally, Moya et al. [4] extended the sampling to include other areas of the Mediterranean Basin and higher latitudes in Europe and found a great number of haplotypes restricted to the Macaronesian region. Altogether, their results suggest that these lichenized fungi probably originated in southern latitudes during relatively recent geological times (Pleistocene) and then expanded north, thus supporting the original hypothesis by Krog and Østhagen [9].

In recent decades, *R. farinacea* has also received considerable attention regarding the diversity and physiology of its associated microalgae (phycobionts) [12]. del Campo et al. [11] and Moya et al. [13] have found that *Trebouxia* Puymaly diversity and the composition in thalli of *R. farinacea* was strongly correlated with the geographic origin of the samples. *Trebouxia jamesii* (Hildreth and Ahmadjian) Gärtner was the main primary phycobiont in thalli from the Iberian Peninsula, while *T. lynnae* Barreno (former *Trebouxia* sp. TR9) was mainly present in thalli from the Canary Islands and Madeira. Furthermore, Casano et al. [14] and del Campo et al. [11] have demonstrated the coexistence of these two *Trebouxia* phycobionts within a single lichen thallus through a combination of microscopic techniques, isolation in axenic cultures, and molecular characterization [5,11]. Casano et al. [14] have found that these two phycobionts exhibited distinct responses to abiotic stresses, and they speculated that their concurrent presence in thalli is advantageous for the holobiont, a phenomenon that has been observed in other lichens with wide distributions and ecologies [15,16,17]. So far, no further studies have attempted to unveil the diversity of the associated *Trebouxia* in *R. farinacea* from areas without the influence of the Mediterranean climate.

The present work provides a global picture of phycobiont diversity and mycobiont–phycobiont interaction patterns in the *R. farinacea* group by considering an enlarged dataset consisting of samples included in Moya et al. [4], newly collected thalli, as well as publicly available sequence data in GenBank. Because previous studies have shown that lichen-forming fungi tend to preferentially select photobiont lineages which are better adapted to local environmental conditions [18,19,20,21,22,23], we will here test the following hypotheses: (i) mycobionts in the *R. farinacea* group associate differentially with phycobionts in Macaronesia and Mediterranean, temperate, and Boreal regions in Europe; and (ii) the association with different phycobionts shapes the ecological distribution of the mycobionts. In doing this, we will investigate the complexity of interactions among mycobiont and phycobiont lineages, the degree of specificity across their geographic distributions [24,25,26], and the climatic niche width of the primary phycobionts.

## 2. Materials and Methods

### 2.1. Sampling, Pretreatment of the Samples, and DNA Extraction

The present study considered 469 thalli of the “*Ramalina farinacea* group”, following the species concepts in Moya et al. [4]: 434 *Ramalina farinacea*, 25 *Ramalina* sp., 5 *R. subfarinacea*, and 5 *R. alisiosae*. Sampling included the Canary Islands and Madeira (Macaronesian archipelagos), the Mediterranean Basin (Algeria, the Iberian and Italian peninsulas, and the Balearic Islands), and central (France, Czech Republic, Austria, and Germany) and northern (Estonia, Finland, Sweden, and Norway) Europe (Figure 1; Appendix A). The sampling localities can be grouped into ten biogeographic subregions, according to Rivas-Martínez et al. [27] (Appendix A): Boreal, Atlantic Europe, Central Europe, Alpine, Mediterranean West and Central Iberian Peninsula, Balearic-Catalonian-Provençal, Italo-Thyrrenian, Adriatic, and Macaronesian. Fresh collected medium size thalli (up to 5 cm) were air-dried and then stored at −20 °C. Before DNA extraction, the thalli were inspected under a stereomicroscope and cleaned with sterile water to avoid contamination by other fungi and epiphytic microalgae. Fragments from different parts of each thallus were arbitrarily excised and pooled together, put into an Eppendorf tube, and ground using a pestle and 400 μL of lysis buffer of the DNeasy Plant Mini kit (Qiagen, Hilden, Germany). Total genomic DNA was extracted and purified using the above kit following the manufacturer’s instructions, and DNA was finally eluted in a volume of 50 μL.

### 2.2. PCR Amplification and Sequencing

The barcode nrITS region was selected to assess the taxonomic identity of the primary phycobiont as well as the mycobiont of the collected *Ramalina* thalli. The primers pairs used were nr-SSU-1780 [29] and ITS4T [30] for the phycobiont, and ITS1F [31] and ITS4a [32] for the mycobiont. PCR reactions were performed in a total volume of 25 μL using the EmeraldAmp GT PCR Master Mix (Takara, Shiga, Japan), which required the addition of the template DNA (1 μL), specific primers (1 μL each, 10 μM), and distilled water. The PCR amplification program comprised an initial denaturation step at 94 °C for 2 min, followed by 30 cycles at 94 °C for 30 s, 56 °C for 45 s, and 72 °C for 1 min, and a final elongation step at 72 °C for 5 min. Amplifications were carried out on 96-well lab cyclers, SensoQuest (Progen Scientific Ltd., Mexborough, UK). The PCR products were then electrophoresed in a 1% agarose gel and visualized using GelRed. The products were purified using the Gel Band Purification Kit (GE Healthcare Life Science, Piscataway, NJ, USA). The amplified PCR products were sequenced with an ABI 3730XL sequencer using the BigDye Terminator 3.1 Cycle Sequencing Kit (Applied Biosystems, Foster City, CA, USA) at StabVida (Lisbon, Portugal). Raw electropherograms were manually checked, trimmed, and assembled using SeqmanII v.5.07© (DnaStar Inc., Madison, WI, USA). GenBank accession numbers are listed in bold in Appendix A (OR978676 to OR979068 for fungal nrITS and OR990615 to OR991088 for *Trebouxia* nrITS).

### 2.3. Inference of Genealogical Relationships among Phycobiont Haplotypes

A phycobiont haplotype network for the *R. farinacea* group was built using an alignment of nrITS sequences. The DNA dataset was assembled with the sequences obtained in the present study and those obtained from the specimens studied in Moya et al. [4] and a selection of 58 sequences from GenBank (available at http://www.ncbi.nlm.nih.gov/; accessed on 19 December 2023) (Appendix A). The search criterion in GenBank was “*Trebouxia* AND *Ramalina farinacea*”, and only those accessions with accompanying information about sampling locality were considered [11,33,34,35]. The final sequence dataset consisted of 527 sequences, which were subsequently aligned with MAFFT v.7.308 [36,37] using the algorithm FFT-NS-I x1000, the 200PAM/k = 2 scoring matrix, a gap open penalty of 1.5, and an offset value of 0.123. The resulting nrITS alignment was manually optimized in Geneious v.9.0.2 to trim the ends of longer sequences that included fragments of 18S–26S ribosomal subunits. Genealogical relationships among haplotypes were then calculated under a statistical parsimony framework in PopART v.1.7 [38] using the TCS method under a 95% parsimony probability criterion, with gaps treated as a 5th character state [39,40,41]. Because the inference of haplotype networks is sensitive to ambiguous base calls and missing data [42], the alignment was first pruned and reduced to 415 complete sequences. The network was artistically edited in Adobe Illustrator 2023 and haplotypes were labelled according to the above-mentioned ten biogeographic subregions [27].

### 2.4. Interaction Networks

We built bipartite interaction networks among phycobiont haplotypes of the *R. farinacea* group and (a) different mycobionts, considering species circumscriptions in Moya et al. [4], (b) mycobiont genetic clusters without any consideration of species boundaries, and (c) biogeographic areas. Hierarchical Bayesian genetic clustering was performed using *fastbaps* v.1.0.8 [43] in R v.4.3.1 [44]. The fast BAPS algorithm is based on applying the hierarchical Bayesian clustering algorithm [45] to the problem of clustering genetic sequences using the same likelihood as BAPS [46]. We employed *optimized.baps* for the Dirichlet prior hyperparameter, as it generally outperforms other prior options [43], and the value of *k.init* was set to 20. This analysis was seeded with a mycobiont nrITS alignment that was built using newly obtained sequences (n = 395) and a selection from GenBank (n = 227) with the following search criteria: “*Ramalina farinacea* AND internal”, “*Ramalina alisiosae* AND internal”, and “*Ramalina subfarinacea* AND internal”. The final dataset consisted of 622 sequences, which were subsequently aligned with MAFFT v.7.308 [36,37] as described above. The three bipartite networks were constructed using the function *plotweb* in the R package *bipartite* [47].

### 2.5. Niche Hypervolumes

Because the two associated microalgae species corresponded to *Trebouxia jamesii* and *T. lynnae* (see Section 3 below), we represented their climatic niche, along with those of the four predominant haplotypes of *T. jamesii*, using the Hutchinsonian niche concept, defining a species’ niche as an n-dimensional hypervolume with dimensions corresponding to environmental variables [48]. The climatic hypervolumes were constructed using multivariate kernel density estimation [49]. A PCA based on 19 bioclimatic variables was conducted [50], and the first two axes, which explained 78% of the total variance, were selected to calculate hypervolumes for each species-level lineage and haplotype. The boundaries of the kernel density estimates were delineated by the probability threshold using the 0.85 quantile value. Hypervolume contours were plotted to project niche spaces based on 5000 random background points, using the alphahull contour type with the alpha smoothing value set to 0.55. For ease of comprehension, differences in climatic variables (BIO1–BIO19) between species were visualized by using box-plots. The significance of a difference between species was tested by using Wilcoxon tests, and Bonferroni correction was applied to adjust the *p*-value. All analyses were performed in R studio v.4.3.1 [44] using base functions and the packages *hypervolume* [49] and *alphahull* [51].

### 2.6. Trebouxia jamesii Geographical Occurrence

The GenBank sequence data of the nrITS were considered to evaluate the global distribution of *Trebouxia jamesii* and the taxonomical breadth of associated mycobionts. BLASTn searches [52] against that database using complete or partial nrITS sequences and following the search criterion “*Trebouxia jamesii* AND internal” were performed; consequently, 1154 hits with a 99–100% of nucleotide identity were retrieved, and their corresponding sequences were downloaded. The final dataset also included four *T. jamesii* haplotypes obtained in the present study and a representative sequence of a strain of *T. jamesii* (FJ626733 UTEX 2233) located at the Culture Collection of Algae of the University of Göttingen, Germany (SAG; available at https://sagdb.uni-goettingen.de/, accessed on 13 January 2024). Sequences of *T. lynnae* were also included in the dataset, and *Asterochloris mediterranea* was used to root the phylogenetic tree. Then, a phylogeny was built to explore within-species phylodiversity and to corroborate the validity of the taxonomic labels of GenBank sequences using an alignment obtained with MAFFT employing the same parameter setting as above. The online version of RAxML-HPC2 hosted at the CIPRES Science Gateway [53,54,55] was used to estimate phylogeny under a maximum likelihood (ML). The analysis used the GTRGAMMA substitution model and one thousand rapid bootstrap pseudoreplicates were conducted to evaluate nodal support. The resulting ML phylogenetic tree was visualized with the iTOL web tool [56], and Adobe Illustrator 2023 was used for artwork. Tree nodes with bootstrap support (BS) values equal to or greater than 70% were regarded as significantly supported. Based on phylogenetic clade clustering, only 211 sequences that matched with representative *T. jamesii* lineages were considered for assessing the geographic distribution and range of associated fungal partners of this microalga (Appendix A) [11,12,33,35,57,58,59,60,61,62,63,64,65,66,67].

## 3. Results

### 3.1. Phycobiont Haplotype Network

The number of nrITS phycobiont haplotypes inferred from the newly sequenced specimens and the selection of phycobiont sequences from the *R. farinacea* group from GenBank was 52. The haplotype network showed the presence of two different *Trebouxia* species, *T. jamesii* and *T. lynnae*, separated by at least 28 different nucleotides (Figure 2). While *T. lynnae* occurred in the Macaronesian archipelagos and the European continent, *T. jamesii* was restricted to continental localities and archipelagos in the Mediterranean Basin. At the level of the 76 sampling localities considered in the present work, *T. jamesii* was the only phycobiont in 60 of them, whereas *T. lynnae* was the sole microalga found in eight localities in the Macaronesian archipelagos and the Iberian and Italian peninsulas (Appendix A). In the eight remaining localities, both phycobiont species co-existed; the predominance of each microalga in these localities varied. Furthermore, considering the distance to the sea from each sampling locality in the straight shortest line (kilometres), *T. jamesii* occurred in both coastal and inland localities (up to 400 km), while *T. lynnae* preferred coastal localities, with a maximum inland occurrence at 58 km in the locality of El Toro in the eastern Iberian Peninsula (Appendix A).

The two most frequent *T. jamesii* haplotypes, H3 and H9, had 21 other haplotypes differing from them by a single nucleotide polymorphism (SNP) (Figure 2). Haplotypes H3 and H9 included specimens from all over the sampling area, except for the Macaronesian archipelagos, while most of the other 21 connected haplotypes were individually represented by a single biogeographical subregion. Haplotypes H1 and H2 were also occurred very frequently in nearly all considered localities. Central Europe and the Mediterranean Central Iberian Peninsula were the two areas hosting a higher number of exclusive *T. jamesii* haplotypes. On the other hand, *T. lynnae* co-occurred with *T. jamesii* in certain localities of the Iberian Peninsula, Balearic Islands, and Italy, which corresponded to the following biogeographic subregions: Atlantic Europe, Mediterranean Central Iberian Peninsula, Balearic-Catalonian-Provençal, and Italo-Thyrrenian. The latter two areas showed the highest number of exclusive minor haplotypes.

### 3.2. fastbaps Cluster Assignment and Interaction Networks

The four species forming the *R. farinacea* group *sensu* Moya et al. [4] showed differential associations to phycobiont species and haplotypes (Figure 3A). *Ramalina farinacea*, *Ramalina* sp., and *R. subfarinacea* were associated with both phycobionts, *T. jamesii* and *T. lynnae*. Only *R. alisiosae* was found to be exclusively associated with *T. lynnae* in Macaronesia (Figure 3A,C). *Ramalina farinacea* specimens (the most abundant in our assembled dataset) formed the highest number of associations with different *T. jamesii* haplotypes. From the phycobiont perspective, the two most-frequent *T. jamesii* haplotypes, H3 and H9, interacted with all mycobiont species except for *R. alisiosae*. The less frequent *T. jamesii* haplotypes (e.g., H6, H19, H20, etc.) only interacted with *R. farinacea*. At a geographical level, we found that both microalgae were not exclusive to a single biogeographic subregion (Figure 3C). Although *T. jamesii* was widespread in continental Europe and present in the Balearic Islands, this species was not detected in Macaronesian samples of the *R. farinacea* group. On the other hand, *T. lynnae* occurred predominantly in biogeographic areas that are often characterized by displaying milder climatic conditions, e.g., Macaronesia, the eastern Iberian Peninsula, and Balearic Islands, as well as localities in Atlantic Europe rather close to the coast. The *Trebouxia jamesii* haplotypes H1, H3, and H9 were shown to be widespread in subregions with markedly different climatic conditions, such as Boreal, Central Europe, Atlantic Europe, and the climatically milder Mediterranean Basin. H3 was also found in North California, which indicates a trans-Atlantic distribution.

Furthermore, the fast BAPS algorithm inferred eight clusters based on the mycobiont nrITS dataset. Figure 3B shows interactions of these clusters (each represented by a Roman number) with phycobiont nrITS haplotypes. For a better comprehension only clusters I, IV, VI, VII, and VIII were represented (Figure 3B). This was because either the remaining clusters derived from mycobiont GenBank nrITS sequences for which there was no information about the associated phycobiont, or they were associated with phycobiont haplotypes consisting of a single sequence/individual. Cluster VIII encompassed the bulk of *R. farinacea* samples and was associated with the highest number of different *T. jamesii* haplotypes. This cluster and the less frequent clusters I, IV, and VI were associated with both *Trebouxia* microalgae. On the opposite side, cluster VII (corresponding to the Macaronesian samples of *R. alisiosae*) was only associated with the three haplotypes of *T. lynnae*.

### 3.3. Climatic Niches

Two-dimensional hypervolumes for *T. jamesii* and *T. lynnae* and for the four most abundant *T. jamesii* haplotypes were built, and PC1 and PC2 explained 78% of the variation in climatic variables (Figure 4). Due to considerable differences in climatic traits among the Macaronesian archipelagos, Mediterranean, and Northern Europe, the hypervolumes of each species were split into two isolated areas, one broader than the other (Figure 4B,C). A discernible gradient of climatic niches along PC2 was observed, explaining variations in temperature and precipitation. Both species exhibited large niches overlapping (Figure 4B) in an area representative of localities placed in the Mediterranean Basin and temperate Europe (especially the Atlantic Europe and Central Europe biogeographic subregions). *Trebouxia jamesii* occurred in more diverse climate types than *T. lynnae*, including an isolated zone in the PCA representing the Boreal European region that shows a wide temperature seasonality. On the other hand, *T. lynnae* showed preference for warmer and drier climates (BIO1, 6, 16, 17, Wilcox tests *p* < 0.001 for all comparisons; Figure 5), and an isolated area related to the Macaronesian region (Figure 4B). The hypervolumes of the most frequent *T. jamesii* haplotypes H1, H2, H3, and H9 revealed rather negligible differences regarding their climatic niches (Figure 4C). H1 and H3 emerged as the most widespread in the analysis, with two distinct areas in the PCA, the smaller representing Boreal Europe.

### 3.4. Phycobiont Geographical Occurrence

A total of 220 sequences were clustered with *T. jamesii* (Appendix A). According to the information reported in GenBank accession metadata, *T. jamesii* has been detected as a primary phycobiont in the following lichen species: *Amandinea punctata*, *Anaptychia runcinata*, *Biatora* sp., *Candelariella vitellina*, *Evernia prunastri*, *Lecanographa amylacea*, *Lecanora argentata*, *L*. *bicincta*, *L*. *chlarotera*, *L*. *frustulosa*, *L*. *glabrata*, *L*. *rupicola*, *L*. *sulphurea*, *Lecidea roseotincta*, *Lecidella elaeochroma*, *Lepra amara*, *Loxospora elatina*, *Melanelixia fuliginosa*, *M. glabratula*, *Myriolecis hagenii*, *Pertusaria amara*, *Pe. coccodes*, *Pe. leioplaca*, *Phlyctis argena*, *Protoparmelia badia*, *Pr. montagnei*, *Pr. psarophana*, *Ramalina calicaris*, *R. capitata*, *R. farinacea*, *R. fastigiata*, *R. fraxinea*, *R. lusitanica*, *R. pollinaria*, *R. pontica*, *Rhizocarpon geographicum*, *Rhizoplaca* sp., *Tephromela atra*, *T. grumosa,* and *Umbilicaria grisea*. According to the analyzed GenBank metadata and our own dataset, the geographic distribution of *T. jamesii* would encompass continental Europe, the British Islands, and islands in the Mediterranean Basin (Austria, Crete, Crimean Peninsula, Czech Republic, Denmark, Estonia, Finland, France, Germany, Greece, Italy, Norway, Poland, Portugal, Slovakia, Slovenia, Spain, Sweden, United Kingdom, and Turkey); Peru in South America and the US (e.g., California) in North America; Algeria in northern Africa; and India in Asia.

Barreno et al. [68] provided a comprehensive analysis of the geographic distribution of *T. lynnae* (see Supplementary Table S2 in [68]), and according to their work and the results here obtained, *T*. *lynnae* associates as a primary phycobiont with *Lecanographa amylacea*, *Protoparmelia montagnei*, the *Ramalina decipiens* group, *R. farinacea*, *R*. *fastigata*, *R. menziesii*, and an undescribed *Ramalina* sp. that belongs to the *R. farinacea* group. Geographically, this microalga is distributed in continental Europe (Italy, Poland, Sweden, and Spain), the Balearic Islands, the Macaronesian archipelagos (Canary Islands and Cape Verde), the island of Madeira, North America, and New Zealand.

## 4. Discussion

Lichens are usually defined as paradigms of mutualistic symbioses, in which fungal mycobionts, populations of photosynthetic microalgae, and other microorganisms interact extracellularly and form morphologically and physiologically complex thalli (holobiomes) [69,70,71,72,73]. A question that still raises considerable curiosity among lichenologists is whether (and why) a given lichenized microalga can be replaced by equally compatible photobionts throughout the lichen’s distribution range [29,64,74,75,76,77,78,79]. This question has been already addressed at various taxonomic, geographic and ecological scales, generating valuable data about the evolutionary history of lichen symbioses [20,21,22,33,80,81,82,83], although, fungal–algal association patterns are still unknown for the vast majority of lichens [73]. In the present study, we explored patterns of symbiont associations in species of the *R. farinacea* group, which comprises *R. subfarinacea*, *R. alisiosae*, and *Ramalina* sp. [4], through an exhaustive European and Macaronesian sampling of specimens spanning ten biogeographic subregions with highly diverse climatic conditions.

Our results revealed that in the sampled regions, the most frequent phycobionts in the *R. farinacea* group were *Trebouxia jamesii* and *T. lynnae*, two phylogenetically closely related species [68]. Associations of closely related mycobionts with single or multiple phycobiont(s) which are also among themselves phylogenetically closely related have been already detected in other lichen symbioses, such as in the Macaronesian *Ramalina decipiens* group [22], the amphitropically distributed *Pseudephebe* spp. [84], or the Mediterranean and temperate epiphytic *Parmelia sulcata* and *P. saxatilis* groups [78]. A pattern is also observed at the level of the single mycobiont species in *Ramalina menziesii* in western North America [33] and the Mediterranean *Seirophora villosa* [85]. These findings do not necessarily mean that strict parallel diversification between lichenized fungi and their green algal partners took place [29,86,87]. Instead, it might suggest that fungal–algal cell recognition mechanisms are rather conserved and that only some interactions succeed during the first lichenization steps. Furthermore, the composition of the “pool” of locally available microalgae may exert a strong influence on the range of possible interactions as well [88,89,90,91]. Singh et al. [35] have found *R. farinacea* in association only with *T. jamesii* in the emblematic Białowieża Forest (Poland). This microalga was the most predominant in the epiphytic lichen community, being present in at least 53% of the sampled species, most of which were phylogenetically unrelated to *R. farinacea* and reproduced either sexually or asexually. Certainly, future research is needed to understand the composition of the “pool” of phycobionts in lichen communities thriving in Mediterranean regions, providing context for our results showing the high diversity of haplotypes and interactions in the *R. farinacea* group.

*Insular* vs. *continental distribution of the phycobionts*—We found that phycobiont association is not balanced amongst the studied mycobionts. The Canarian *R. alisiosae* showed a strict association with *T. lynnae*, whereas the other three mycobionts interacted with both *T. jamesii* and *T. lynnae*. Although it might be questioned that the results are biased by the low number of samples of *R. alisiosae* against those of *R. farinacea*, our results are well in line with those of Blázquez et al. [22], who have used high-throughput sequencing techniques to evaluate the phycobiont diversity in fifteen species of the *Ramalina decipiens* group in the Canary Islands, Madeira, and Cape Verde. Although the authors did not find remarkable differences in the identity of associated phycobionts, *T. lynnae* turned out to be the most frequent microalgal partner. The less frequent phycobionts were *T. jamesii*, *T. aggregata*, *T. decolorans*, *T. australis,* and some still undescribed *Trebouxia* species. The *R. decipiens* group represents a putative radiation of endemic lichenized fungi inhabiting the Macaronesian region, whose diversification does not seem to be linked with differing associations to phycobiont lineages [22]. The *R. farinacea* group likely diversified in the Macaronesian–Mediterranean region as well [4,9]. The strict interaction of *R. alisiosae* with *T. lynnae* is in line with the observation that this microalga is better adapted to the environment conditions of oceanic islands than other phycobionts [12,14,18,22]. This is further supported by the lack of *T. jamesii* lineages in the Macaronesian thalli of *R. farinacea*. However, the phylogeny of the *R. farinacea* group is still not well resolved [4], and due to the low number of studied *R. alisiosae* specimens we cannot assess whether phycobiont switch could have contributed to diversification in our target species.

The relationship of the lichen *R. farinacea* with *T. jamesii* breaks in the Canary Islands, Madeira, Sicily, and the Balearic Islands, where the mycobiont also (or only) associates with *T. lynnae*. del Campo et al. [11] and Molins et al. [12] have previously suggested a preference for continentality for *T. jamesii* and insularity for *T. lynnae*, which is further evidenced by our ecological niche analysis. The preference for warmer and more humid climates of *T. lynnae* aligns with the results of Casano et al. [14], who have demonstrated that this microalga has better photosynthetic performance at higher temperature and irradiance, while *T. jamesii* thrives at moderate temperature and irradiance. The better physiological performance of *T. lynnae* under more oxidative conditions than *T. jamesii* may reflect its greater capacity to undertake key metabolic adjustments, including increased non-photochemical quenching, higher antioxidant protection, and the induction of repair mechanisms [14]. These considerations align well with the previous physiological analyses performed on axenic isolates in vitro of the two phycobionts.

Furthermore, Hinojosa-Vidal et al. [92] and Pérez-Rodrigo et al. [93] have studied the effects of prolonged exposure to high salt concentrations on both *T. lynnae* and *T. jamesii* species and have demonstrated the extraordinarily higher tolerance to osmotic and saline stress of these two *Trebouxia* species compared to other microalga genera, such as *Asterochloris* and *Chlorella*. Their results suggest that the two *Trebouxia* phycobionts could cope with highly saline environments. Thus, a clear relationship was observed in the present study between *T. lynnae* occurrence and the distance to the sea, which might be related to its tolerance to salinity by water sprays and aerosols. Although *T. jamesii* also inhabits coastal areas (which could also be explained by its tolerance to salinity), it is able to colonize inland areas where *T. lynnae* does not spread into. Coastal zones usually display a warmer climate due to the isothermal influence of the air’s relative humidity, and temperatures tend to be more stable throughout the year. Contrary, inland zones experience more pronounced seasonal thermal variations. It is likely that a combined effect of the influence of temperature and distance from the sea controls the distribution of these microalgae. Even within thalli, the tolerances and physiological roles of photobionts vary; for example, two different photobionts provide different polysaccharides under changing seawater immersion in the marine lichen *Lichina pygmaea* [79]. Thus, it is a good example for testing hypotheses of how complex photobiont communities might react to environmental changes in other lichen symbioses, such as the cases studied in this work.

*The multiplicity of haplotypes*—Previous studies have already described a specific association of *T. jamesii* and *T. lynnae* with *R. farinacea* based only on Iberian Peninsula and Canary Islands specimens [11,12,14]. Here, we found that the association between *R. farinacea* and *T. jamesii* spanned from 36 to 63 degrees of latitude, i.e., from northern Africa (Algeria) to Central, Alpine, and Boreal regions in continental Europe. Some phycobiont haplotypes were restricted to localities in Central and Boreal Europe (e.g., H1, H6, H19, and H35), which might point towards fine-tuned fungal–algal interactions in these regions, assuming the existence of “pools” of potentially better-adapted microalgae [88,89,90,94]. In fact, phycobiont switches have been suggested for various epiphytic, terricolous, and saxicolous lichens throughout their distribution ranges, often aligning with notable changes in environmental conditions [18,20,61,81,84,85,94,95,96]. Alternatively, rare haplotypes might just represent the genetic differentiation in the phycobiont by intrathalline somatic mutations, as suggested for the microalgae occurring in the lichens *Xanthoria parietina* and *Anaptychia ciliaris* [97]. However, there were other *T. jamesii* haplotypes, such as H3 and H9, which showed a widespread distribution across different biogeographic regions. This finding is not strange, as identical nrITS phycobiont haplotypes have been reported in symbiosis with the same mycobiont over large geographic distances, even encompassing temperate, north, and south Polar Regions [84,94]. Given that *R. farinacea* reproduces mainly asexually using soredia, in which both symbionts are dispersed together, we might hypothesize that this lichen achieved its current broad geographic distribution by a successful first co-dispersion. Secondly, once the soredium developed and hyphae grew further, the mycobiont would have settled in that area by enwrapping those locally adapted phycobionts, i.e., those that were found to be less frequent and more geographically restricted haplotypes. This switch might have been achieved in at least two ways in *R. farinacea*: by the association of fungal hyphae projecting from soredia with a new phycobiont, as suggested by Nelsen and Gargas [86] in the strictly asexually reproducing *Lepraria*; alternatively, by a de novo lichenization processes in which case the mycobiont would reproduce sexually through the dispersion of meiotic spores. The number of different geographically adapted photobiont haplotypes may also be influenced by the species richness of the lichen communities present at a certain site. Events of photobiont switches are known to occur in lichens [17,23,29,65], especially if mycobionts and photobionts present low specificity and selectivity toward themselves, and thus many phycobionts are available for lichenization on growth substrates (i.e., wood, rocks, soil).

The present study has extended our knowledge about the geographic distribution and range of possible mycobiont partners of the two studied phycobionts. *Trebouxia lynnae* associates with lichen fungi belonging to phylogenetically distant families and from different areas of the planet, including continental Europe (Iberian Peninsula, Poland, and Sweden), the Balearic Islands, Macaronesian archipelagos (Canary Islands, Madeira, and Cape Verde), North America, and New Zealand [22,68]. *Trebouxia jamesii* is also a cosmopolitan phycobiont, which associates with several lichenized fungi [35,98].

## 5. Conclusions

In this research, we faced certain shortcomings when using the GenBank database to retrieve phycobionts sequences using nrITS BLASTn searches. Indeed, we found that out of the 1154 nrITS sequences labelled as *T. jamesii* in GenBank, only 220 corresponded to the authentic *T. jamesii*. This confusion might have stemmed from having formerly considered *T. jamesii* and *T. simplex* conspecific taxa due to the lack of morphological differences [99]. Nonetheless, differences between these two phycobionts were later found and the distinction of both species was finally accepted [63,89,100,101]. It is currently acknowledged that the authentic (type) strain of *T. jamesii* (UTEX 2233) belongs to the *Trebouxia* clade A [102], whereas *T. simplex* falls within the *Trebouxia* clade S. It is thus crucial to emphasize that species identification of *Trebouxia* cannot rely solely on BLASTn matches. In any study involving symbiotic microalgae, it is a necessary condition that identifications should be based on phylogenies encompassing species from all clades, including the 27 species officially described to date and indicated as references for phylogenetic datasets [103].

Future studies must reveal whether the apparently strict association of mycobiont members in the *R. farinacea* group with *T. jamesii* and *T. lynnae* is maintained in other areas of the planet, especially in North America. It would also be essential to examine whether the different haplotypes of *T. jamesii* found across its European distribution perform differently at the physiological level, and if the differences are related to varying environmental conditions.

## Figures and Tables

**Figure 1 jof-10-00206-f001:**
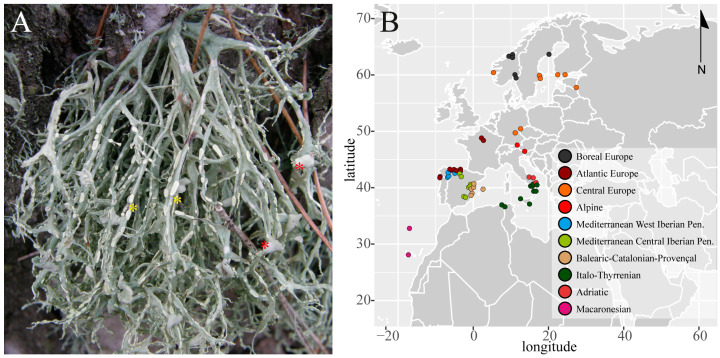
(**A**) An epiphytic thallus of *Ramalina farinacea* collected in Bocairent (eastern Spain; photo: IGB) showing soredia (yellow asterisks) and apothecia (red asterisk). (**B**) Georeferenced map showing sampling localities of specimens of the “*R. farinacea* group” in the ten studied biogeographic subregions: Boreal (n = 25), Atlantic Europe (n = 56), Central (n = 51) Europe, Alpine (n = 6), Mediterranean West (n = 11), Central (n = 167) Iberian Peninsula, Balearic-Catalonian-Provençal (n = 35), Italo-Thyrrenian (n = 52), Adriatic (n = 14), and Macaronesian (n = 6). The map was created using the function *map_data* in the R package *ggplot2* [28].

**Figure 2 jof-10-00206-f002:**
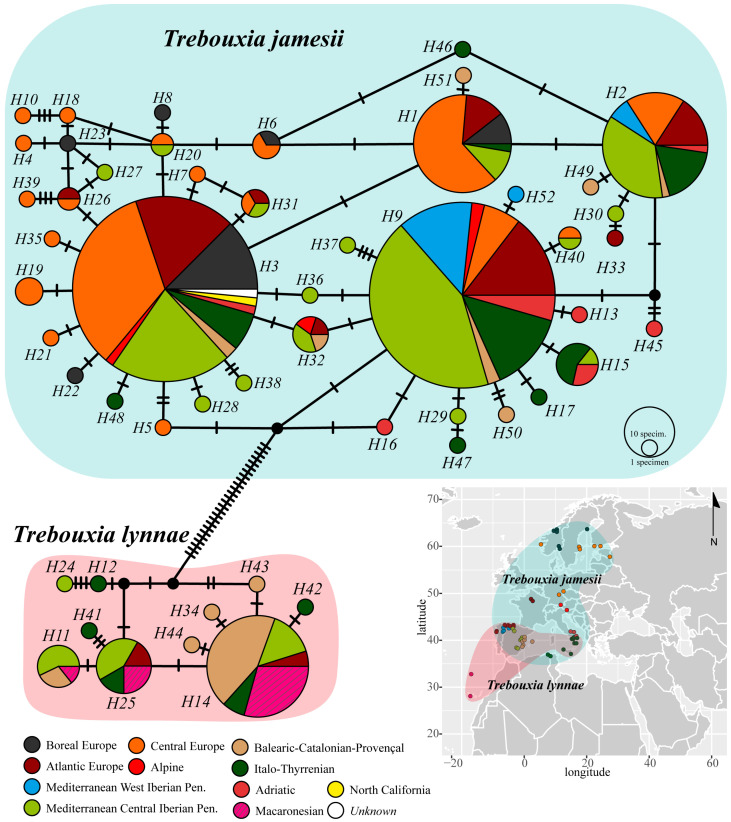
Statistical parsimony network of phycobiont haplotypes calculated using the extended nrITS sequence sampling that included available GenBank data of *Trebouxia* from *Ramalina farinacea*; circle colors indicate the biogeographic subregions where specimens were collected (see legend below); the sizes of circles in each network are proportional to the numbers of individuals bearing the haplotype; circles may represent two or more haplotypes when these are separated only by indels; black-filled circles indicate missing haplotypes, and hatch marks indicate mutations. The geographic distributions of the two phycobionts of the *R. farinacea* group are represented over the map.

**Figure 3 jof-10-00206-f003:**
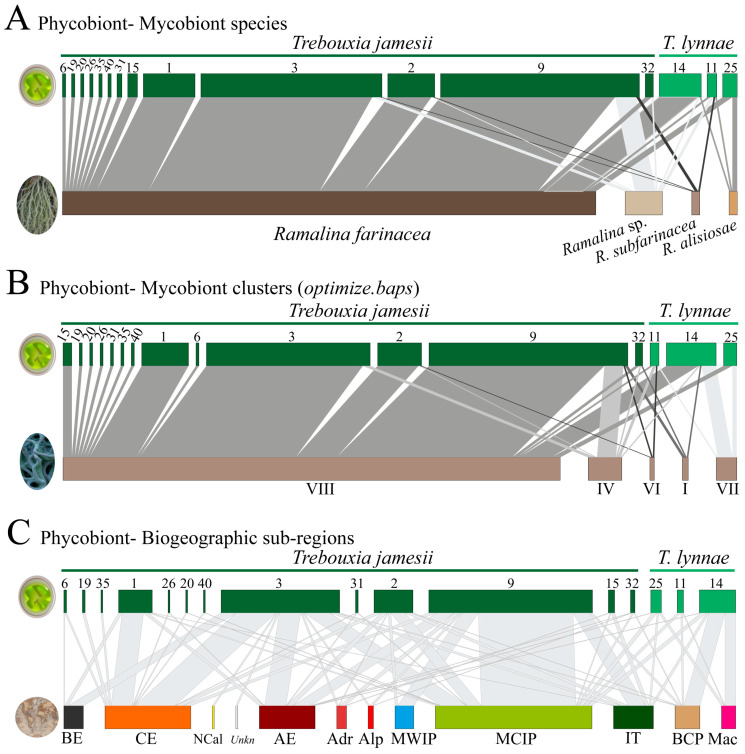
Interaction networks among the phycobionts *Trebouxia jamesii* and *T. lynnae* and (**A**) the different mycobiont species in the “*R. farinacea* group”; (**B**) mycobiont genetic clusters delimited with *fastbaps* (I, IV, VI, VII and VIII); and (**C**) biogeographic subregions spanning from Macaronesian archipelagos to continental Europe. The width of the links is proportional to the number of specimens forming the association. The size of rectangles depicting each phycobiont haplotype is proportional to the number of obtained sequences; however, the sizes are not proportional among the three networks. Roman numerals represent different genetic clusters inferred with *fastbaps*. The biogeographic subregions [27] where specimens were collected are indicated as BE: Boreal Europe, CE: Central Europe, NCal: North California, *Unkn*: Unknown; AE: Atlantic Europe, Adr: Adriatic, Alp: Alpine, MWIP: Mediterranean West Iberian Peninsula, MCIP: Mediterranean Central Iberian Peninsula, IT: Italo-Thyrrenian, BCP: Balearic-Catalonian-Provençal, Mac: Macaronesian.

**Figure 4 jof-10-00206-f004:**
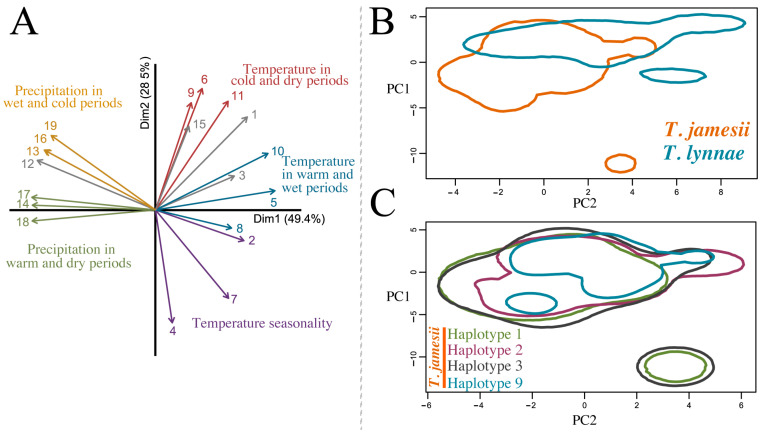
(**A**) Principal coordinate analysis (PCA) of the 19 BioClim variables: 1 = annual mean temperature, 2 = mean diurnal range, 3 = isothermality, 4 = temperature seasonality, 5 = max temperature of warmest month, 6 = min temperature of coldest month, 7 = temperature annual range, 8 = mean temperature of wettest quarter, 9 = mean temperature of driest quarter, 10 = mean temperature of warmest quarter, 11 = mean temperature of coldest quarter, 12 = annual precipitation, 13 = precipitation of wettest month, 14 = precipitation of driest month, 15 = precipitation seasonality, 16 = precipitation of wettest quarter, 17 = precipitation of driest quarter, 18 = precipitation of warmest quarter, 19 = precipitation of coldest quarter. Climatic niche hypervolumes based on climatic PC1–PC2 axes (78% of variation explained) for (**B**) phycobiont species *T. jamesii* and *T. lynnae* and for (**C**) the four most abundant *T. jamesii* haplotypes.

**Figure 5 jof-10-00206-f005:**
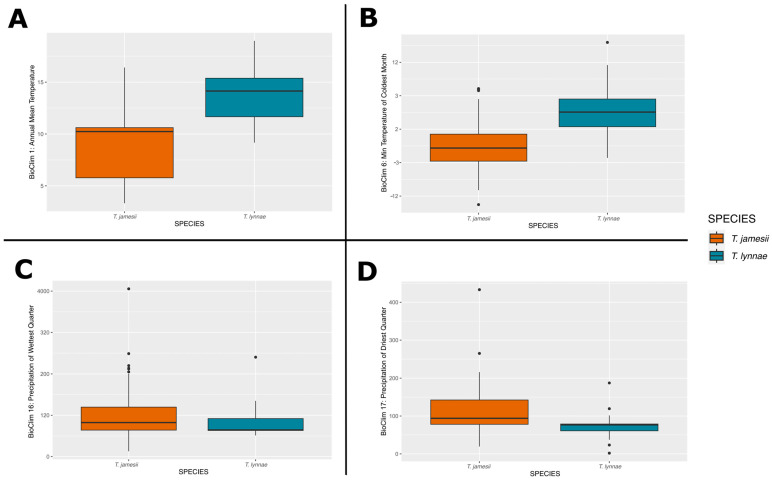
Box-plot diagram representing differences in climate preferences in *Trebouxia jamesii* and *T. lynnae*. Climatic data were obtained from the Global Climate Data—WorldClim. (**A**) BIO1 = annual mean temperature (°C), (**B**) BIO6 = min temperature of coldest month (°C), (**C**) BIO16 = precipitation of wettest quarter and (**D**) BIO17 = precipitation of driest quarter.

## Data Availability

The dataset generated during the current study is available in the GenBank (see Appendix A): OR978676 to OR979068 for fungal nrITS and OR990615 to OR991088 for *Trebouxia* nrITS.

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
