# Peer review of "Myco–Phycobiont Interactions within the “Ramalina farinacea Group”: A Geographical Survey over Europe and Macaronesia"

_jof, 2024, doi:10.3390/jof10030206_

Round 1

Reviewer 1 Report

In the manuscript “A geographic overview of myco-phycobiont symbiotic interatction within the Ramalina farinacea group”, the authors analyse a collection of samples from across Europe for both fungal and algal identity, describing patterns of association. They apply a suite of climate variables to look for explanations about the patterns observed. The authors choose to emphasize a climatic adaptation model for patterns of association between R. farinacea (and relatives) partner with particular algae.

The paper is well written, and the sampling in Europe is wide with good descriptive figures describing and displaying the main results. Of the over 500 sequenced samples, 68 include the rarer and more restricted algal species Trebouxia lynnae, with the rest associating with T. jamesii, reflecting the depth of the sampling. The paper represents a lot of work and a lot of good analysis, and it is a substantial advance on the state of knowledge of this system.

I have three main issues that deserve further thought. First, it would be instructive to test hypotheses relating to the distributions of algae, rather than presenting only descriptive analysis; the conclusions are sometimes presented without reference to other potential explanations of the data. For example, the overall conclusion that T. lynnae is more restricted to warmer and more humid situations, corroborating earlier evidence, can be tested by comparing the appropriate climate variables relating to an appropriate hypothesis rather than showing all variables. There is a very wide overlap of niche space, but it is very likely possible to demonstrate different trends or tolerances between the two species using only the variables of interest. This would be a stronger conclusion than only showing the hypervolumes.

Second, the possibility of a different model entirely for the photobiont associations is never presented, i.e. the idea that locally abundant algae in the microhabitats of the lichens being studied may be associated more frequently purely due to their availability rather than to local adaptation. This is hardly addressed.

Third, near the end of the manuscript, after a lot of work to explain the climatic adaptation model, we find that the ‘specialist’ alga T. lynnae is found well outside the climate envelope discussed at length in the paper. See comment for L470.

I have a number of specific suggestions that I hope will improve the manuscript
Title: Should focus on Europe, not suggest that the paper is a geographic overview of the entire species complex throughout the world.

L35. Please use a simpler word rather than laciniae. If you feel you need to define soredia, then you would certainly need to define this or just use a simple term like ‘lobe’ or ‘branch’.

L39. Delete ‘Furthermore’ and alter to read “R. farinacea is distributed widely…”

L72. Alter to read “…that Trebouxia Puymaly diversity and composition in thalli…”

L93. Distribution misspelled

L96. I do not see a direct link between the findings of this study and evolutionary history of lichen symbioses.

L112. See comment above L35; arbitrary does not equal random; if there is useful information from prior studies about the distribution of different algal partners along the length of a lichen thallus reflecting age differences, please insert that here, e.g. Molins et al. 2021.

L141. The numbers should be listed here (e.g. ZYxxxxxx-ZYxxxxxx, GHxxxxxx-GHxxxxxx).

L171. The analysis was ‘seeded’? ‘fed’? ‘started’?

L184-187. For readers not familiar with these methods, please briefly outline the data fed into these analyses, i.e. is the value for each climate variable for each specimen used? Without this, it is difficult to interpret the differences in estimated hypervolumes, particularly in Fig4c.

Section 2.6 is missing italics for scientific names

L226. A map showing the different algal species in each sampling location would be better than a list, though the list is also needed.

L235. Cite Figure 2, but these don’t show a star-like shape. I know what you mean, but maybe just say that they are related to many haplotypes differing by a single-mutational step.

Figure 2. It is pretty tricky to tell Alpine from Macronesia in the colour scheme. Could you use hatching to show those types most closely associate with T. lynnae for clarity? This would also help show where they are found in T. jamesii.

L256-257. The interpretation is slightly confounded with the presentation of results here. Please remove ‘because’. It is very likely true, but you can’t answer that so unequivocally.

L272-274. Awkward construction

L280. Word choice: indistinctly seems the wrong word here

Figure 3. Hard to see ‘ja’ and ‘ly’ in the legend.

L310. H1 and H3 are among the most numerically abundant types, but interesting that H9 is the most numerically abundant but slightly more limited in niche. Hmmm.

L340. Algeria misspelled.

L346. Should read “…this microalga is distributed…”

L354. I would say that thalli being considered unites of evolution is probably fairly contentious, as the partners so readily separate. Interesting to ponder, but a distraction here, and not clearly drawn as relevant to any other of your points.

L366. Somewhere, and maybe in the intro and again here, the actual range of the species/species complex should be mentioned. Europe is a small part of what may be a cosmopolitan distribution.

L400-402. I don’t think there is good enough evidence to show what the available pool of photobionts is in any of these areas, so trying to say that there is a ‘prioritization’ or local adaptation rather than just an association based on differential abundance within the locally available pool is not possible. Consider the scenario where T. lynnae may more abundant within R. farinacea s lat. because it is also the photobiont of many other co-occurring epiphytes: the pattern would be the same either way and the speculation about local adaptation only makes sense in light of the physiological work of Casano and colleagues. Just a caution to not overstate conclusions when other explanations have not been examined.

L432. Usually it is the thermoregulation provided by the sea that is used to explain coastal isothermality. Maybe it is just awkward wording here?

L436-9. Awkward wording. Can the point of this be brought out more clearly?

L450 Notable, not notorious.

L470. Well, this is a very big surprise if T. lynnae is also in Poland, Sweden, undefined parts of North America and New Zealand?! This is directly counter to quite a lot of this discussion and suggestion that local adaptation to warmer and wetter temperatures. This goes back to the that it is not the abundance of a photobiont in a site that is important, but rather the combination of fungal and algal bionts together that seems to be behind the geographic patterning.

L472. Replace ‘worldwide distributed’ with ‘cosmopolitan’ or say ‘distributed worldwide’

L 481. Delete “, which”.

Acknowledgements: L534: should read “for contributing collections and Lucie …”

Last, I missed two bits of information to add context: 1) a short discussion – even a sentence about how the results of this analysis using only the dominant type of alga fit within the context of knowing that more algae are often present within a thallus. 2) the definition of T. lynnae corresponding to TR9, to ensure the older literature can be mapped onto these findings.

Author Response

First, it would be instructive to test hypotheses relating to the distributions of algae, rather than presenting only descriptive analysis; the conclusions are sometimes presented without reference to other potential explanations of the data. For example, the overall conclusion that T. lynnae is more restricted to warmer and more humid situations, corroborating earlier evidence, can be tested by comparing the appropriate climate variables relating to an appropriate hypothesis rather than showing all variables. There is a very wide overlap of niche space, but it is very likely possible to demonstrate different trends or tolerances between the two species using only the variables of interest. This would be a stronger conclusion than only showing the hypervolumes.

It is indeed true that ecological niche analyses can sometimes be challenging to interpret, as noted by the reviewer, particularly when there is significant overlap between the niches of the two species. In order to clarify our findings and highlight the specific climatic variables that support the hypothesis that T. lynnae is tends to prefer warmer conditions, we have supplemented Figure 4 (see new Figure in Supplementary Material) with a representation in the form of box-plot diagrams of different variables that show a clear difference in climatic preferences of T. jamesii vs T. lynnae.

Second, the possibility of a different model entirely for the photobiont associations is never presented, i.e. the idea that locally abundant algae in the microhabitats of the lichens being studied may be associated more frequently purely due to their availability rather than to local adaptation. This is hardly addressed.

Thank you for the suggestion. We have attempted to introduce other possible explanations for the model found into the text.

Third, near the end of the manuscript, after a lot of work to explain the climatic adaptation model, we find that the ‘specialist’ alga T. lynnae is found well outside the climate envelope discussed at length in the paper. See comment for L470.

See answer in L470

Detail comments

I have a number of specific suggestions that I hope will improve the manuscript
Title: Should focus on Europe, not suggest that the paper is a geographic overview of the entire species complex throughout the world.

While it is true that our sampling is predominantly European, it also includes samples from Northern Africa and Macaronesia. Perhaps the term "geographic overview" is somewhat vague and does not specify the exact area. However, at no point did we intend to provide a geographic overview of the entire distribution of the RF complex. To address this potential confusion, we have modified the title to: " Myco-phycobiont interactions within the “Ramalina farinacea group”: a geographical survey over Europe and Macaronesia."

L35. Please use a simpler word rather than laciniae. If you feel you need to define soredia, then you would certainly need to define this or just use a simple term like ‘lobe’ or ‘branch’.

We have modified the phrase to read as 'whitish-greenish fruticose thalli composed by slender laciniae (thallus ramifications), as indicated on line 36. This formulation to explain the term 'laciniae' was previously used in Moya et al., 2017.

L39. Delete ‘Furthermore’ and alter to read “R. farinacea is distributed widely…”

Done

L72. Alter to read “…that Trebouxia Puymaly diversity and composition in thalli…”

Done

L93. Distribution misspelled

Done

L96. I do not see a direct link between the findings of this study and evolutionary history of lichen symbioses.

The phrase "evolutionary history" has been removed from the text, and the paragraph has been modified. Please refer to line 100 for the changes.

L112. See comment above L35; arbitrary does not equal random; if there is useful information from prior studies about the distribution of different algal partners along the length of a lichen thallus reflecting age differences, please insert that here, e.g. Molins et al. 2021.

The material and methods section has been revised, specifically in lines 112 and 115, to avoid using the term "lacinia" and to better explain how the starting material for extraction was obtained. Following Molins et al. 2021, particularly when working with RF, an attempt is made to mix parts of the thallus and not work with very small thalli (up to 1 cm long). Additionally, whenever possible, intermediate-sized thalli are uniformly utilized.

L141. The numbers should be listed here (e.g. ZYxxxxxx-ZYxxxxxx, GHxxxxxx-GHxxxxxx).

Done

L171. The analysis was ‘seeded’? ‘fed’? ‘started’?

Rephased to seeded.

L184-187. For readers not familiar with these methods, please briefly outline the data fed into these analyses, i.e. is the value for each climate variable for each specimen used? Without this, it is difficult to interpret the differences in estimated hypervolumes, particularly in Fig4c.

With the purpose of generating a climatic niche hypervolume, the software retrieves the values of each independent climatic variable, including annual temperature, mean precipitation, etc., for each coordinate. Following this, a Principal Component Analysis (PCA) is conducted using the complete dataset, encompassing data of all 19 climatic variables obtained for each coordinate. This PCA estimates which climatic variables carry the most weight in determining the climatic niche.

This methodology has been previously used in several lichen studies:

https://onlinelibrary.wiley.com/doi/full/10.1111/mec.14764

https://nsojournals.onlinelibrary.wiley.com/doi/full/10.1111/ecog.03457

https://www.frontiersin.org/journals/microbiology/articles/10.3389/fmicb.2021.769304/full

Nonetheless, should further elucidation be necessary, we could contemplate integrating the graphical representation illustrating the distribution of variables (Fig 4a) superimposed onto each hypervolume figure, although this may potentially difficult image visualization.

Section 2.6 is missing italics for scientific names

Done, thank you.

L226. A map showing the different algal species in each sampling location would be better than a list, though the list is also needed.

In order to clarify this result and supplement the information provided in Table S3, we have now included a map in Figure 2 with the requested data. Thank you for the suggestion

L235. Cite Figure 2, but these don’t show a star-like shape. I know what you mean, but maybe just say that they are related to many haplotypes differing by a single-mutational step.

Done

Figure 2. It is pretty tricky to tell Alpine from Macronesia in the colour scheme. Could you use hatching to show those types most closely associate with T. lynnae for clarity? This would also help show where they are found in T. jamesii.

With the aim of facilitating the differentiation between the Alpine and Macaronesian regions, we have added hatching the the colors in Figure 1 and 2. Regarding the second question, we have included a new Figure (Map) with the distribution of the two algae.

L256-257. The interpretation is slightly confounded with the presentation of results here. Please remove ‘because’. It is very likely true, but you can’t answer that so unequivocally.

Rephrased.

L272-274. Awkward construction

Rephrased.

L280. Word choice: indistinctly seems the wrong word here

Word indistinctly deleted

Figure 3. Hard to see ‘ja’ and ‘ly’ in the legend.

We have modified the whole Figure 3 and now we do not use “ja” and “ly”, so the figure has become more understandable.

L310. H1 and H3 are among the most numerically abundant types, but interesting that H9 is the most numerically abundant but slightly more limited in niche. Hmmm.

The main difference between H1, H3, and H9 is that the first two are located in the Boreal region, whereas H9 is not in the Boreal region but rather in the Mediterranean West Iberian Peninsula. This difference in geographic distribution results in distinct niche distributions.

L340. Algeria misspelled.

Done.

L346. Should read “…this microalga is distributed…”

Done.

L354. I would say that thalli being considered unites of evolution is probably fairly contentious, as the partners so readily separate. Interesting to ponder, but a distraction here, and not clearly drawn as relevant to any other of your points.

Sentence deleted

L366. Somewhere, and maybe in the intro and again here, the actual range of the species/species complex should be mentioned. Europe is a small part of what may be a cosmopolitan distribution.

To clarify this matter, several modifications have been made: the title, in the Introduction, and  in the Discussion. We hope that these adjustments are sufficient.

L400-402. I don’t think there is good enough evidence to show what the available pool of photobionts is in any of these areas, so trying to say that there is a ‘prioritization’ or local adaptation rather than just an association based on differential abundance within the locally available pool is not possible. Consider the scenario where T. lynnae may more abundant within R. farinacea s lat. because it is also the photobiont of many other co-occurring epiphytes: the pattern would be the same either way and the speculation about local adaptation only makes sense in light of the physiological work of Casano and colleagues. Just a caution to not overstate conclusions when other explanations have not been examined.

Thank you for the suggestion. We have attempted to introduce other possible explanations for the model found into the text.

L432. Usually it is the thermoregulation provided by the sea that is used to explain coastal isothermality. Maybe it is just awkward wording here?

Done.

L436-9. Awkward wording. Can the point of this be brought out more clearly?

We rephrased the sentence to clarify, but it the referee still consider it awkward we can delete it: On another scale of geographic distances between populations, Chrismas et al. [79] studied within-habitat variation in the thalli of the two cyanobionts of the marine lichen Lichina pygmaea (Lightf.) C. Agardh in relation to fluctuating conditions of the intertidal zone. They proved that Rivularia and Pleurocapsa produced different solute polysaccharides to be processed extracellularly in carbon acquisition by the mycobiont being in relation to seawater immersion. Thus, it is a good example for testing hypotheses on how complex photobiont communities might react to environmental changes in other lichen symbioses, such as the cases studied in this work.

L450 Notable, not notorious.

Done.

L470. Well, this is a very big surprise if T. lynnae is also in Poland, Sweden, undefined parts of North America and New Zealand?! This is directly counter to quite a lot of this discussion and suggestion that local adaptation to warmer and wetter temperatures. This goes back to the that it is not the abundance of a photobiont in a site that is important, but rather the combination of fungal and algal bionts together that seems to be behind the geographic patterning.

In the case of the North American locality, it is California, which meets the climatic conditions defended in this study. As for New Zealand, the researchers who uploaded the sequence to GenBank did not provide reference to the locality or the island. It could be a temperate locality. Only in the case of Poland, which is in the Bialowieza National Park, and Sweden, which is in Uppland, it would contradict our hypothesis. However, by introducing other possibilities that explain this pattern in the Discussion, we refrain to do further changes.

L472. Replace ‘worldwide distributed’ with ‘cosmopolitan’ or say ‘distributed worldwide’

Done.

L 481. Delete “, which”.

Done.

Acknowledgements: L534: should read “for contributing collections and Lucie …”

Done.

Last, I missed two bits of information to add context:

  • a short discussion – even a sentence about how the results of this analysis using only the dominant type of alga fit within the context of knowing that more algae are often present within a thallus.

In the Discussion, it is reported that Del Campo et al. [11] and Molins et al. [12] previously suggested a preference for continentality and insularity in T. jamesii and T. lynnae, respectively, which is further evidenced by our ecological niche analysis of the two phycobionts. Molins et al. [12] used metabarcoding analyses to corroborate the presence of more than one Trebouxia within an individual thallus. Our results, using only Sanger sequencing, are consistent with the findings of Molins et al. for the predominant alga. However, regarding the other algae present in each thallus, we cannot make any conclusions because the techniques are not comparable.

 2) the definition of T. lynnae corresponding to TR9, to ensure the older literature can be mapped onto these findings.

The paper describing T. lynnae, published in 2022, clearly refers to it as the older TR9. Nonetheless, the first time T. lynnae is mentioned in the introduction, we have added "(former Trebouxia sp. TR9)."

Reviewer 2 Report

The paper represents a comprehensive investigation of phycobiont diversity within “R. farinacea group”. The study is well written and will be interesting for a broad auditorium. The data analysis was adequate. Authors were careful in their suggestions and made a good literature review. I have only general suggestions to make the ms more attracting and easier for the understanding. I suggest to start the Introduction with the description of the global problem(s), which the study helps to solve. Is it lichen mechanisms of evolution or interaction with phycobionts, or response to different environments... choose something and put this from the beginning. Then I suggest to move to Ramalina farinacea description, but tell in the beginning that it is a complex of species.

The Discussion may be divided in several paragraphs with their own titles, this helps to comprehend the text.

I do not have any necessarily detail corrections.

Author Response

The paper represents a comprehensive investigation of phycobiont diversity within“R. farinácea group”. The study is well written and will be interesting for a broad auditorium. The data analysis was adequate. Authors were careful in their suggestions and made a good literature review. I have only general suggestions to make the ms more attracting and easier for the understanding. I suggest to start the Introduction with the description of the global problem(s), which the study helps to solve. Is it lichen mechanisms of evolution or interaction with phycobionts, or response to different environments... choose something and put this from the beginning. Then I suggest to move to Ramalina farinacea description, but tell in the beginning that it is a complex of species.

We thank the reviewer for this suggestion, still would like to decline it and do not change the paragraphs in the Introduction. We think that mentioning only about the species complex of Ramalina, would be inappropriate without describing the species itself. We have smoothed a bit the species description and improved the Introduction in its last paragraph according to the other Reviewers’ suggestions. We hope the Reviewer would be happy with.

The Discussion may be divided in several paragraphs with their own titles, this helps to comprehend the text.

DONE. We improved a bit the entire text of the Discussion and have subdivided it into smaller sections with own title to ease the reading. In doing this, we separated the last paragraphs into the section “Conclusions”.

Reviewer 3 Report

1. In our opinion, is the selected DNA barcode (nrITS) more successful than the rbcL gene that has been used for members of the order Trebouxiales (for example: Skaloud et al., 2015; Vancurova et al., 2015; Garrido-Benavent et al., 2022)? What advantages or disadvantages does it have?

Skaloud, P., Steinová, J., Rídká, T., Vancurová, L. & Peksa. O. (2015). Assembling the challenging puzzle of algal biodiversity: species delimitation within the genus Asterochloris (Trebouxiophyceae, Chlorophyta). 51(3): 507-527.

Vancurova, L., Peksa, O., Nemcová, Y. & Skaloud, P. (2015). Vulcanochloris (Trebouxiales, Trebouxiophyceae), a new genus of lichen photobiont from La Palma, Canary Islands, Spain. Phytotaxa 291: 118-132.

Garrido-Benavent, I., Chiva, S., Bordenave, C.D., Molins A. & Barreno, E. (2022). Trebouxia maresiae sp. nov. (Trebouxiophyceae, Chlorophyta), a new lichenized species of microalga found in coastal environments. Cryptogamie, Algologie 43(9): 135-145.

2. P12L502: “representative sequence of a strain of T. jamesii …” Do you mean the authentic strain SAG 2103? Please indicate it in the text.

3. It would be helpful if you plotted the number of lichen samples collected in the legend of Figure 1B for each of the 10 subregions studied.

4. Please note that the name Trebouxia lynnae probably does not follow standard rules of orthography and has been renamed to Trebouxia lynniae (https://www.algaebase.org/search/species/detail/?species_id=185501).

5. What genetic differences of nrITS were observed between T. jamesii and T. lynniae? Are they interspecific or intraspecific for the genus Trebouxia? 28 different nucleotides is this the threshold for species separation?

6. Could environmental conditions have influenced the smaller cell size of T. lynniae compared to T. jamesii? And which ones do you think?

7. What do you think about phorophyte? Is there a relationship between it and a phycobiont? Is there any data on the acidity of the phorophyte bark?

8. Can ITS1 and ITS2 secondary structures contribute to valid species delimitation for Trebouxia? Is the CBC approach sensu A. Coleman successful for this algal genus?

Coleman AW (2000) The significance of a coincidence between evolutionary landmarks found in mating affinity and a DNA sequence. Protist 151: 1–9

Coleman AW (2009) Is there a molecular key to the level of "biological species" in eukaryotes? A DNA guide. Mol Phylogenet Evol 50: 197–203

9. Currently, lichens are considered holobionts, do you plan to use a metagenomic approach in future studies to study all components of the lichen?

1. P2L93: distributio -> distribution

2. P3L121: Mediterranian -> Mediterranean

3. P6L262: were not exclusive of a single -> were not exclusive to a single

4. P6L265, P7L281: On the opposite -> On the opposite side

5. P7L281: corresponding the Macaronesian -> corresponding to the Macaronesian

6. P7L282: only associated -> is only associated

7. P10L427: Their results suggests -> Their results suggest

8. P11L488: are kept -> is kept

Author Response

Major comments

  1. In our opinion, is the selected DNA barcode (nrITS) more successful than the rbcL gene that has been used for members of the order Trebouxiales (for example: Skaloud et al., 2015; Vancurova et al., 2015; Garrido-Benavent et al., 2022)? What advantages or disadvantages does it have?

Skaloud, P., Steinová, J., Rídká, T., Vancurová, L. & Peksa. O. (2015). Assembling the challenging puzzle of algal biodiversity: species delimitation within the genus Asterochloris (Trebouxiophyceae, Chlorophyta). 51(3): 507-527.

Vancurova, L., Peksa, O., Nemcová, Y. & Skaloud, P. (2015). Vulcanochloris (Trebouxiales, Trebouxiophyceae), a new genus of lichen photobiont from La Palma, Canary Islands, Spain. Phytotaxa 291: 118-132.

Garrido-Benavent, I., Chiva, S., Bordenave, C.D., Molins A. & Barreno, E. (2022). Trebouxia maresiae sp. nov. (Trebouxiophyceae, Chlorophyta), a new lichenized species of microalga found in coastal environments. Cryptogamie, Algologie 43(9): 135-145.

The choice of a molecular marker depends on various factors, including the organism under study, the purpose of the study, and the availability of data in databases for that marker to enable comparisons. In the case of describing a new species of Trebouxiales, particularly Trebouxia, the rbcL marker, along with ITS and cox2, has been established by Muggia et al. (2020) as necessary markers for inferring robust phylogenies. However, due to the abundance of information available in GenBank, ITS as marker performs well in diversity analyses, including phylogenies. In fact, it is also considered a good barcode for Trebouxiales.

  1. P12L502: “representative sequence of a strain of T. jamesii…” Do you mean the authentic strain SAG 2103? Please indicate it in the text.

It has been specified in the Material and Methods section, and it is also highlighted in bold in the phylogenetic tree of Figure S3.

  1. It would be helpful if you plotted the number of lichen samples collected in the legend of Figure 1B for each of the 10 sub-regions studied.

Done

  1. Please note that the name Trebouxia lynnaeprobably does not follow standard rules of orthography and has been renamed to Trebouxia lynniae (https://www.algaebase.org/search/species/detail/?species_id=185501).

As suggested by the reviewer, we recently became aware that the species name has been modified (without prior consultation with the authors) on AlgaeBase. It appears that according to Article 60.8: -iae is the feminine ending for a name ending in a consonant, thus lynniae. However, this modification has not been formally implemented, and although the correct name is likely to be lynniae, we cannot accept this change unless it is officially made. Since the species was formally described in 2022 as T. lynnae, we have decided for now to maintain this name.

  1. What genetic differences of nrITS were observed between T. jamesiiand T. lynniae? Are they interspecific or intraspecific for the genus Trebouxia? “28 different nucleotides” is this the threshold for species separation?

The genetic differences observed only in the ITS region for both species are more than 40 nucleotides. They are clearly two distinct species.

  1. Could environmental conditions have influenced the smaller cell size of T. lynniaecompared to T. jamesii? And which ones do you think?

In other studies, such as the description of T. lynnae, a comparative table of the morphological and ultrastructural traits of Trebouxia species on clade A is provided in Table 1. In this table, T. lynnae is reported to have slightly smaller cell size compared to T. jamesii. These measurements were likely obtained from cells in culture rather than in symbiosis. To address this question, an ultrastructural study comparing thalli freshly collected from symbiotic associations in areas with different climatic conditions would be necessary.

  1. What do you think about phorophyte? Is there a relationship between it and a phycobiont? Is there any data on the acidity of the phorophyte bark?

It is true that the acidity of the host bark can significantly influence the pool of available algae; however, it does not seem to be a determining factor in this study.

  1. Can ITS1 and ITS2 secondary structures contribute to valid species delimitation for Trebouxia? Is the CBC approach sensu A. Coleman successful for this algal genus?

Coleman AW (2000) The significance of a coincidence between evolutionary landmarks found in mating affinity and a DNA sequence. Protist 151: 1–9

Coleman AW (2009) Is there a molecular key to the level of "biological species" in eukaryotes? A DNA guide. Mol Phylogenet Evol 50: 197–203

Indeed, the study of secondary structures would likely provide significant results in the delineation of species within the genus Trebouxia. However, that is not the objective of this study.

  1. Currently, lichens are considered holobionts, do you plan to use a metagenomic approach in future studies to study all components of the lichen?

We have previously conducted metabarcoding studies with RF (Moya et al., 2017 and Molins et al., 2021). As part of the current project, it is planned to perform a metabarcoding analysis of some of the samples included in this Sanger-only analysis.

Detail comments

  1. P2L93: distributio -> distribution

Done.

  1. P3L121: Mediterranian -> Mediterranean

Done.

  1. P6L262: were not exclusive of a single -> were not exclusive to a single

Done.

  1. P6L265, P7L281: On the opposite -> On the opposite side

Done.

  1. P7L281: corresponding the Macaronesian -> corresponding to the Macaronesian

Done

  1. P7L282: only associated -> is only associated

Done.

  1. P10L427: Their results suggests -> Their results suggest

Done.

  1. P11L488: are kept -> is kept

Done.

Round 2

Reviewer 1 Report

No major comments; most comments are very minor having to do with textual suggestions. 

L28 should read "...to show the association with different..."

L78 phycobiont misspelled

L89 "research" or "studies"

L91 please don't say you prove, but you will test or provide support for hypotheses

Fig 2 The map distributions are really helpful and nicely presented

Fig 3 is much easier to read and comprehend. I would recommend removing the little symbols on the left, though. The algae symbol/icon makes sense, but the ones for the fungi aren't obvious and distract from the clarity of the data presented here.

L317-318 With lots of tests, make sure you adjust the p-value accordingly (Bonferoni correction). If you already know what variables you're going to test, just do those.

L392 may exert (not exerts)

L397 Reword "Certainly, future research is needed to ..."

L398 "... providing context for our results showing high diversity..." Is that what you mean?

L414 I don't follow this.

L417 maybe the observation rather than the fact

L429 performance

L452 showed, not proved

L450-54 This needs reworded at least. That paper show different physiological roles for the different photobionts along with the carbon sources from each and how they are probably processed and used by the fungi. Certainly cool, but I don't see the link immediately as written. You could say something like: Even within thalli, tolerances and physiological roles of photobionts vary; for example, two different photobionts provide carbon differentially under different conditions in the marine lichen Lichina pygmaea (Chrismas et al. 2024).

L489 Be specific here: geographically adapted photobiont haplotypes

Author Response

L28 should read "...to show the association with different..."

Done

L78 phycobiont misspelled

Done

L89 "research" or "studies"

Done

L91 please don't say you prove, but you will test or provide support for hypotheses

Done

Fig 2 The map distributions are really helpful and nicely presented

Thank you for the recommendation

Fig 3 is much easier to read and comprehend. I would recommend removing the little symbols on the left, though. The algae symbol/icon makes sense, but the ones for the fungi aren't obvious and distract from the clarity of the data presented here.

In this case, we do not consider it necessary to change the figure since we believe that the symbols of the algae, fungi or map do provide clarity. If possible, we would like to keep it as it is.

L317-318 With lots of tests, make sure you adjust the p-value accordingly (Bonferoni correction). If you already know what variables you're going to test, just do those.

The Bonferroni adjustment for the p-value was already added for the Wilcox test. However, this information had not been written in the text; we have now pointed it out.  On the other hand, for the niche analysis, all variables are used (each with different weight). However, in box-plots we have only represented the most significant variables.

L392 may exert (not exerts)

Done

L397 Reword "Certainly, future research is needed to ..."

Done

L398 "... providing context for our results showing high diversity..." Is that what you mean?

We have reworded the sentence: providing context for our results showing high diversity of haplotypes and interactions in the R. farinacea group.

L414 I don't follow this.

We have reworded the sentence: The R. decipiens group represents a putative radiation of endemic lichenized fungi inhabiting the Macaronesian region, whose diversification does not seem to be linked with differing associations to phycobiont lineages.

L417 maybe the observation rather than the fact

Done

L429 performance

Done

L452 showed, not proved

Done

L450-54 This needs reworded at least. That paper show different physiological roles for the different photobionts along with the carbon sources from each and how they are probably processed and used by the fungi. Certainly cool, but I don't see the link immediately as written. You could say something like: Even within thalli, tolerances and physiological roles of photobionts vary; for example, two different photobionts provide carbon differentially under different conditions in the marine lichen Lichina pygmaea (Chrismas et al. 2024).

We have tried to reworded it following your suggestions

L489 Be specific here: geographically adapted photobiont haplotypes

Done